# Epigenomic Alterations of the Human *CYP11B* Gene in Adrenal Zonation

**DOI:** 10.3390/ijms252211956

**Published:** 2024-11-07

**Authors:** Yoshimichi Takeda, Masashi Demura, Takashi Yoneda, Shigehiro Karashima, Mitsuhiro Kometani, Daisuke Aono, Seigo Konishi, Shin-ichi Horike, Yasuhiro Nakamura, Yuto Yamazaki, Hironobu Sasano, Yoshiyu Takeda

**Affiliations:** 1Saiseikai Kanazawa Hospital, 13-6 Akadochyo-ni, Kanazawa 920-0353, Japan; aldo_takeda@yahoo.co.jp; 2Department of Health Promotion and Medicine of Future, Graduate School of Medical Science, Kanazawa University, 13-1 Takara-machi, Kanazawa 920-8641, Japanskarashima@staff.kanazawa-u.ac.jp (S.K.); kometankomekome@yahoo.co.jp (M.K.); daisukeaono2007@yahoo.co.jp (D.A.); seigo524-_-728@hotmail.co.jp (S.K.); 3Department of Hygiene, Graduate School of Medical Science, Kanazawa University, 13-1 Takara-machi, Kanazawa 920-8640, Japan; m-demura@med.kanazawa-u.ac.jp; 4Division of Functional Genomics, Research Center for Experimental Modeling of Human Disease, 13-1 Takara-machi, Kanazawa 920-8640, Japan; sihorike@staff.kanazawa-u.ac.jp; 5Division of Pathology, Tohoku Medical and Pharmaceutical University 1-15-1 Fukumuro, Miyaginoku, Sendai 983-8536, Japan; yasu-naka@tohoku-mpu.ac.jp; 6Department of Pathology, Tohoku University Hospital, 2-1 Seiryo-machi, Aoba-ku, Sendai 980-8575, Japan; y.yamazaki@patholo2.med.tohoku.ac.jp (Y.Y.); hsasano@patholo2.med.tohoku.ac.jp (H.S.); 7Hypertension Center, Asanogawa General Hospital, 83 Kosakamachi-naka, Kanazawa 920-8621, Japan

**Keywords:** DNA methylation, adrenal medulla, *CYP11B2*, *CYP11B1*, adrenal zonation

## Abstract

The *CYP11B2* gene is sporadically expressed in the zona glomerulosa (ZG), whereas the *CYP11B1* gene is detected in the zona fasciculata (ZF)/reticularis (ZR), with predominant expression in the ZF. We studied the association between DNA methylation and adrenal zonation. Next, the *CYP11B2* methylation statuses in the adrenal medulla (n = 4) and pheochromocytomas (n = 7) were examined. The expression of *CYP11B2* in pheochromocytomas and non-functioning adenomas (NFAs) (n = 4) was also studied. Adrenals from five autopsy subjects were assessed for immunohistochemically defined adrenal zonation. We used laser capture microscopy to isolate DNA from each zone in adrenal tissues. *CYP11B1* was predominantly unmethylated in the ZF but heavily methylated in the ZG and the ZR. In contrast, *CYP11B2* was hypomethylated in the ZG compared with in the ZF and the ZR. In terms of the expression site and strength, the promoter methylation patterns for *CYP11B2* and *CYP11B1* showed capacities to express CYP11B enzymes. The DNA methylation patterns of the *CYP11B2* and *CYP11B1* promoters were closely associated with adrenal zonation. The unmethylated CpGs of *CYP11B2* were found in the adrenal medulla and pheochromocytomas. Gene expression of *CYP11B2* was detected in the pheochromocytomas. These results indicate the possibility that the synthesis of aldosterone occurs in the adrenal medulla. Further study is necessary to elucidate the pathophysiological roles for the synthesis of aldosterone in the adrenal medulla.

## 1. Introduction

The human adrenal gland is divided into three main zones: the glomerulosa, the fasciculata, and the reticularis. Each has a distinct function, producing aldosterone, cortisol, and androgens, respectively [1,2]. The glomerulosa steroidogenic function (the expression of *CYP11B2*) and the final stage of aldosterone synthesis are induced by angiotensin II stimulation via the angiotensin type 1 receptor and potassium [3,4]. The fasciculata, which is ACTH-dependent, is modulated by the melanocortin type 2 receptor (MC2R) and acts through cAMP formation [5,6].

Epigenetic changes are inherited modifications that are not part of the DNA sequence. Gene expression is regulated at various levels, including via DNA modifications. Of these modifications, histone acetylation regulates gene expression, and DNA hypermethylation induces gene silencing. Gene expression is also regulated by RNA modifications, which mediate RNA metabolism [7,8].

The DNA methylation at the 5′-cytosine of CpG dinucleotides is a major epigenetic modification in eukaryotic genomes and is required for mammalian development. There is a negative correlation between the mRNA levels and methylation rates of *CYP11B2* in aldosterone-producing adenomas [9,10]. We previously reported the epigenetic control of *CYP11B2*, in which case human adrenal cells did not produce aldosterone by the complete methylation of the *CYP11B2* promotor region. Adrenal cells transfected with partially methylated *CYP11B2* decreased the *CYP11B2* expression and aldosterone synthesis compared to those transfected with the unmethylated gene [10].

Cortisol biosynthesis is mainly regulated by the cyclic AMP (cAMP)/protein kinase A (PKA) signaling pathway activated by ACTH secreted from the anterior pituitary gland [11]. In this pathway, 11β-hydroxylase (CYP11B1) catalyzes the final step of cortisol biosynthesis [12]. In cortisol-producing adenomas, Kometani et al. [5] found that the promoter region of *CYP11B1* was hypomethylated. They also reported that cAMP decreased the methylation rate of *CYP11B1* and increased the *CYP11B1* expression in human adrenal cells. There is growing evidence that epigenetics is a regulatory layer that may contribute to the adrenal-zone-specific pattern of enzyme expression. In humans, CYP17A1 plays an important role in cortisol biosynthesis, while, in rodents, 3β-HSD is important for corticosterone biosynthesis. CpG islands are reported to be present in rodents but absent in humans, and methylation and gene expression are reported to be related in rodents [13,14]. The aim of this study was to evaluate the association between the DNA methylation of *CYP11B* genes and adrenal zonation in humans.

We have reported a patient with a pheochromocytoma associated with the overproduction of aldosterone. The *CYP11B2* gene expression was increased and the methylation status of *CYP11B2* was low in this patient [15]. The simultaneous occurrence of primary aldosteronism and a pheochromocytoma is generally considered to be extremely rare [16]. This is because these two lesions arise from different origins. However, Mai et al. [17] reported the mutation of the potassium channel KCNJ5 in the pheochromocytoma with hyperaldosteronism. These results might suggest that chromaffin cells could produce aldosterone. We analyzed the methylation status of the *CYP11B2* genes in the adrenal medulla, pheochromocytomas, and non-functioning adrenal adenomas (NFAs).

## 2. Results

### 2.1. The CpG Methylation Status of the CYP11B2 Promoter Region in Zones of the Human Adrenal Cortex

By using an LCM technique, the CpG methylation status was examined in the three adrenal gland zones of five autopsy specimens. The CpG methylation at CpG1 (Ad1) and CpG2 (Ad5) was hypo- or normo-methylated in the ZG compared with that of the ZF in all five subjects. Inter-individual diversity was observed in the DNA methylation at CpG3 (Figure 1). CpGs 1, 2, and 3 were hypomethylated in the ZG, whereas CpG2 was hypermethylated in the ZF. The methylation ratios of CpGs1, 2, and 3, on average, were low, in the order of ZG, ZF, and ZR (Figure 1).

### 2.2. CpG Methylation Status of the CYP11B1 Promoter Region in Zones of Human Adrenal Cortex

The *CYP11B1* (CpG1-3) methylation ratio was very low in the ZF. Both the ZG and the ZR showed hypermethylation of the *CYP11B1* gene (Figure 1).

### 2.3. CpG Methylation Status and Expression of the CYP11B2 Gene in the Human Adrenal Medulla, Pheochromocytomas, and Non-Functioning Adenomas

*CYP11B2* mRNAs were detected in three samples of pheochromocytomas and NFAs. We could not measure the *CYP11B2* mRNAs of the adrenal medulla because of the paraffin-embedded tissue. Four samples of pheochromocytomas were completely methylated. Three samples of pheochromocytomas, NFAs, and adrenal medullas showed partial methylation (Figure 2).

## 3. Discussion

The three zones of the human adrenal cortex are functionally distinct, with the ZG producing aldosterone, the ZF producing cortisol, and the ZR producing DHEA/DHEA-sulfate. This functional differentiation is largely due to the zone-specific expression of steroidogenic enzymes. In this study, we found DNA hypomethylation at CpGs 1 and 2 in the ZG and hypermethylation at CpG2 in the ZF. We have reported that the methylation of CpG1 greatly decreased the CREB1 binding to Ad1. The DNA methylation at CpG2 reduced the basal binding activities of NR4A1 (NGFIB) and NR4A2 (NURR1), respectively [10]. Meanwhile, the DNA methylation increased the MECP2 binding to CpG1 and CpG2. Lu et al. [18] showed that both NGFIB and NURR1 immunoreactivities are abundant in the ZG and the ZF in contrast to the ZR. Taken together, the patterns of CpG methylation and expression levels of NGFIB and NURR1 appear to produce a synergistic effect on the *CYP11B2* transcription in the adrenal cortex. Thus, hypomethylation at CpGs 1 and 2 in the ZG and hypermethylation at CpG2 in the ZF are likely important for the zone-specific expression of *CYP11B2*.

The ZR was generally hypermethylated. The trans-acting elements involved in *CYP11B2* gene transcription may be lacking in the ZR. In fact, NGFIB and NURR1 are expressed at very low levels in the ZR compared to in the ZG or the ZF (20). Irrespective of the DNA methylation status, *CYP11B2* gene expression is absent in the ZR, indicating that the DNA methylation of the *CYP11B2* promoter in the ZR is less important for *CYP11B2* gene expression. Thus, the CpG methylation of CpGs 1 and 2 may play a role in establishing the functional zonation in the human adrenal cortex by regulating *CYP11B2*. Tezuka et al. [19] analyzed sixty adrenal glands from deceased kidney transplant donors and from patients with renal cancer who underwent en bloc nephrectomy and adrenalectomy procedures and found that functional AG (CYP11B2-positive) areas declined with aging in men but not in women. van de Wiel et al., [20] also reported a negative correlation between age and the relative CYP11B2 expression area in human adrenals from birth to 40 years of age. The number of our samples is small. More samples are necessary to determine the relationship between *CPP11B2* methylation and gender difference or aging.

Cortisol-producing adenoma (CPA) expresses *CYP11B1* but not *CYP11B2* [21,22,23]. Kubota-Nakayama et al. [24] reported that the gene and protein expression of *CYP11B1* were increased in CPAs. We reported that higher mRNA levels of *CYP11B1* were associated with a lower methylation ratio in CPAs compared to adrenal tissues or non-functioning adenomas [5]. The CpG dinucleotides in the *CYP11B1* promoter were found to be exclusively unmethylated in the ZF but not in the ZG and the ZR. This is consistent with the ZF-specific expression of *CYP11B1*.

In our study, the quantification of the mRNA of *CYP11B* in each adrenal zone was not conducted because we could not obtain sufficient RNA for the q-PCR. The next-generation sequencing technologies for human adrenal-zone-specific microRNA and DNA methylation profiling should be necessary for understanding the epinegetics in adrenal zonation.

Duparc et al. [25] reported that CYP11B2-positive cells were present within the adrenal medulla in close contact with the chromaffin cells. Ugi et al. [16] reported the coexistence of a pheochromocytoma and primary aldosteronism due to multiple aldosterone-producing mictonodules in the ipsilateral adrenal gland. However, there have been no reports that chromaffin cells synthesize aldosterone. The mRNA of the StAR gene, *CYP11A*, 3β-hydroxysteroid dehydrogenase, *CYP21*, *CYP11B1*, and *CYP11B2* are expressed in blood vessels and the heart [3,10,26,27]. We found that the *CYP11B2* mRNA levels were lower in the renal arteries than in the adrenal gland. Moreover, hypermethylation was observed in the renal arteries [10]. Briones et al. [28] reported that the expression of the aldosterone synthase gene and protein were detected in 3T3-L1 cells and mature adipocytes, which produce aldosterone basally and in response to angiotensin II. In 3T3-L1 adipocytes, angiotensin II increased the expression of *CYP11B2*.

Aldosterone synthesis both in the central nervous system and the peripheral nerves has been reported [27,29]. Mohamed et al. [30] reported that aldosterone immunoreactivity, *CYP11B2* gene expression, and the MR protein were abundant in the peptidergic nociceptive neurons of dorsal root ganglia. Dehe et al. [31] identified mineralocorticoid receptors, aldosterone, and its processing enzyme CYP11B2 on parasympathetic and sympathetic neurons in intracardiac ganglia. We detected *CYP11B2* gene expression in pheochromocytomas. The *CYP11B2* mRNA levels and the methylation status in some pheochromocytomas were almost equal with those of NFAs. There are many reports of potassium channel KCNJ-5 mutation in primary aldosteronism [32,33,34]. Several KCNJ5 mutations induce the upregulation of *CYP11B2* expression and aldosterone synthesis, leading to increased aldosterone production. Mai et al. [17] reported the interesting case of a pheochromocytoma with hyperaldosteronism in which focal proliferation of enlarged medullary-like cells was observed and mutations of KCNJ5 were detected. Although we did not measure the CYP11B2 mRNA of the adrenal medulla, the adrenal medulla might be able to synthesize aldosterone. Further study is necessary to clarify the pathological roles of aldosterone synthesis in the adrenal medulla and pheochromocytomas.

In summary, in terms of the expression site and strength, the promoter methylation patterns of *CYP11B2* and *CYP11B1* reflect the capacity to express CYP11B enzymes. The DNA methylation patterns of the *CYP11B2* and *CYP11B1* promoters were closely associated with adrenal zonation. The adrenal medulla and pheochromocytomas might be able to synthesize aldosterone.

## 4. Materials and Methods

### 4.1. Human Tissue Collection

Human adrenal glands were obtained during autopsies performed at Kanazawa University Hospital (Kanazawa, Japan) and Tohoku University Hospital (Sendai, Japan). The purpose of the study was explained, and written informed consent was obtained from all study participants. The use of these tissues was approved by the Human Genome/Gene Analysis Research Ethics Committee of Kanazawa University (No. 208-10, 28 March 2019) and Tohoku University School of Medicine Ethics Committee (2022-1-978, 21 February 2023).

Adrenal tumors were collected after removal by surgery. Patients with pheochromocytoma (43 ± 9 years old, urinary metanephrine 0.8 ± 0.6 mg/day, urinary normetanephrine 9.0 ± 1.3 mg/day, plasma aldosterone concentration 131 ± 8 pg/mL, and plasma renin activity 1.0 ± 0.9 ng/mL/h) were diagnosed according to the guidelines of the Endocrine Society [35]. Non-functioning adrenal adenomas (NFAs) (n = 4) were found incidentally by computed tomography scan for unrelated reasons. Patients with clinical NFA had no signs or symptoms of hormone excess, normal serum potassium levels, and plasma cortisol levels suppressible with 1 mg of dexamethasone. All samples were frozen in liquid nitrogen and stored at −80 °C. Both DNA and RNA were simultaneously isolated using an ISOGEN (NIPPON GENE Co., Ltd., Tokyo, Japan) and used for analyses of CpG methylation status and mRNA expression, respectively.

### 4.2. Laser Capture Microdissection

The human adrenal cortex is composed of 3 distinct zones: the zona glomerulosa (ZG), the zona fasciculata (ZF), and the zona reticularis (ZR). These 3 zones have functionally distinct roles in adrenocortical steroid hormone production. We used laser capture microscopy (LCM) to isolate DNA from each zone in the adrenal tissues and it was analyzed for CpG methylation status. Immunohistochemistry was used to distinguish the ZG, the ZF, and the ZR using antibodies for 17alpha-hydroxylase/17,20-lyase (CYP17), 3beta-hydroxysteroid dehydrogenase (HSD3B), and dehydroepiandrosterone (DHEA)-sulfotransferase (SULT2A1), respectively, as previously reported [18,36]. CYP17, HSD3B, and SULT2A1 are expressed exclusively in the ZF/ZR, ZG/ZF, and ZR, respectively.

Five adrenal tissue cases were cryosectioned at a thickness of 10 μm. Each adrenal cortex zone was laser-transferred from the adrenal tissue under light microscopic examination. For LCM/DNA methylation analysis, genomic DNA was isolated using a QIAamp DNA micro kit (QIAGEN K.K.).

Four human adrenal medullas were macroscopically separated.

### 4.3. DNA Methylation Assay

Genomic DNA was treated with bisulfite and amplified. Human *CYP11B2* and *CYP11B1* promoter regions were amplified using specific primers according to a previous report (Table 1) [4,7]. The CpG sites within the *CYP11B2* and *CYP11B1* promoter regions are shown in Figure 3. Bisulfite sequencing was performed using the Methylation DNA Modification Kit (EPIGENTEK, Farmingdale, NY, USA) with specific primers (Table 1). Eight PCR product clones generated from the DNA of each sample were picked to analyze the DNA methylation status, as we described previously [3]. Methylation analysis by pyrosequencing was also performed using PyroMarkGold Q96 Reagents and the PyroMarkQ24 pyrosequencing system (Qiagen, Germantown, NJ, USA) [10].

### 4.4. Real-Time Reverse Transcriptase PCR

Real-time PCR for human *CYP11B2* was carried out using the TaqMan Gene Expression Assay (Applied Biosystems, Life Technologies Japan Ltd., Tokyo, Japan). Real-time PCR for internal controls was carried out using the SYBR Green method. The primer sequences were previously reported [10]. Triplicates of each cDNA sample and no template controls were added to each real-time PCR run.

### 4.5. Statistics

Statistically significant differences were determined by the two-tailed Student’s or Welch’s *t*-test. *p* values less than 0.05 were considered statistically significant.

## Figures and Tables

**Figure 1 ijms-25-11956-f001:**
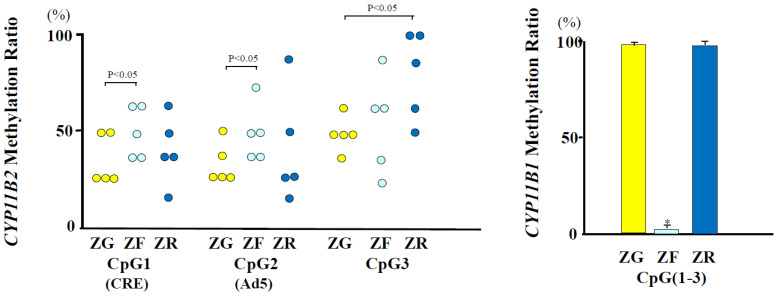
Association of the total methylation ratios of *CYPB* promoters with adrenal zonation. The methylation ratio of CpG1 or CpG2 of the *CYP11B2* in the ZG was significantly lowered compared to that in the ZF (*p* < 0.05). The methylation ratio of CpG3 in the ZG was significantly decreased compared with the ZR (*p* < 0.05). *CYP11B1* promoters (CpG1-3) showed very low methylation ratios in the ZF. Each DNA clone was observed in Appendix A. ZG, zona glomerulosa (yellow color), ZF, zona fasciculata (light blue color), ZR, zona reticularis (blue color); *: *p* < 0.001 vs. ZG, ZF; data of *CYP11B1* methylation are presented in Appendix A.

**Figure 2 ijms-25-11956-f002:**
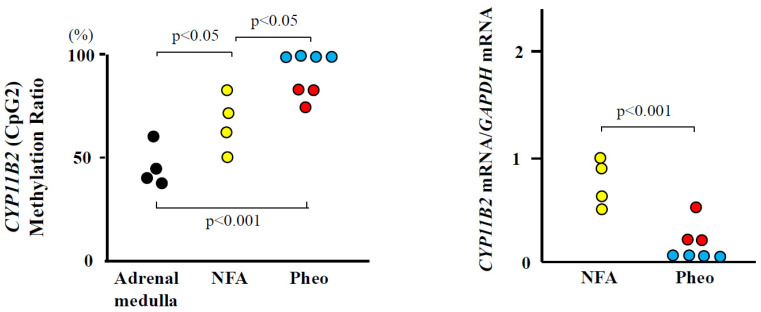
*CYP11B2* methylation ratio (CpG1) in the adrenal medulla was significantly lowered compared with non-functioning adrenal adenomas (NFAs) (*p* < 0.05) or pheochromocytomas (Pheos) (*p* < 0.001). Totally methylated *CYP11B2* promoters were observed in four cases of Pheos. *CYP11B2* gene expression in the NFA was significantly increased compared with Pheos (*p* < 0.001). Four cases (blue circles) were not detectable.

**Figure 3 ijms-25-11956-f003:**
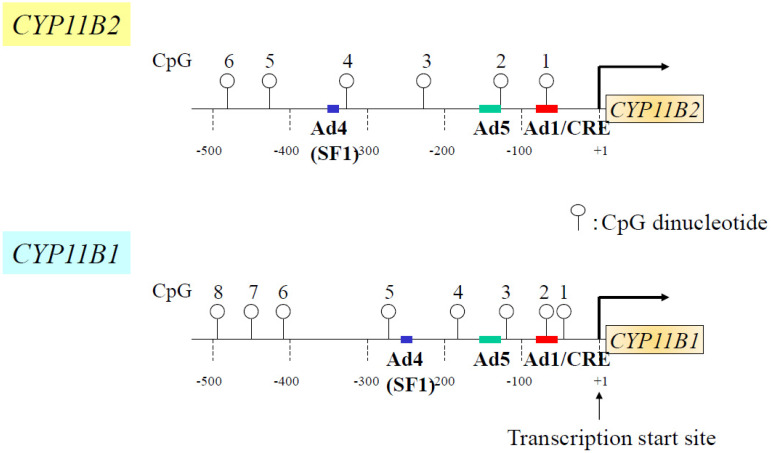
Schema of CpG dinucleotides within the human *CYP11B2* and *CYP11B1* gene promoters. Open circle denotes CpG dinucleotides. Ad, cis-acting element; SF-1, steroidogenic factor-1; CRE, cyclic AMP responsive element.

**Table 1 ijms-25-11956-t001:** Oligonucleotide DNA used in this study.

Pyrosequence	
Primer	Sequence (5′ to 3′)
*CYP11B1* Pyro-F 1-2	TTGTAATTTTTTTATTTTGTTTGGTGTTT
*CYP11B1* Pyro-R 1-2	ATACACCCCCAATAAATCCCTAC
*CYP11B1* Pyro-S1	TGTTTGGTGTTTTGTTTT
*CYP11B1* Pyro-S2	TGGTTTTGGATTTGTTTGAG
*CYP11B1* Pyro-F3	AGGTTAGGGTTGGAGGTAGG
*CYP11B1* Pyro-R3	AACCCCATCCATCTTACTCCTC
*CYP11B1* Pyro-S3	ATTGGGGGTGTATGA
*CYP11B1* Pyro-F4	GGATGGGGTTTTTATTTTATTTAAGAGT
*CYP11B1* Pyro-R4	CCCAATAATCATTCAAAAACAAATTACTCA
*CYP11B1* Pyro-S4	ATTTATTTTTTTGTAAGGTTTATA
*CYPB2* Pyro-F1	TTTTATTTAGGAATTTGTTTTGGAAATATA
*CYPB2* Pyro-R1	AAACACCTAACTTCTCCTTCATCTAC
*CYPB2* Pyro-S1	AGGAATTTGTTTTGGAAATATATTA
*CYPB2* Pyro-F2	ATTGGTTTTGGATTTGTTTGAGATT
*CYPB2* Pyro-S2	GGATTTGTTTGAGATTTTTAGA
*CYPB2* Pyro-F3	GTTGGAGGTTTTTAGTTAAAGGTAGAT
*CYPB2* Pyro-R2	TTCCACCAACATAAACCCCCAAT
*CYPB2* PyroS3	GTTTGAGGATGTTGAGA
Bisulfite sequencing	
Human *CYP11B2*F1	GAAAGGAGAGGTTAGGTTTTATTATTTT
Human *CYP11B2*R1	ACCCTTTACAAAAACAACCAAAA
Human *CYP11B2*F2	GAAAGGAGAGGTTAGGTTTTATTATTTT
Human *CYP11B2*R2	ACCCTTTACAAAAACAACCAAAA
Real-time reverse transcriptase PCR	
*GAPDH* (F)	TCATTGACCTCAACTACATGGTTT
*GAPDH* (R)	TTGATTTTGGAGGGATCTCG

‘F’, ‘R’, and ‘S’ indicate forward, reverse, and sequence primers, respectively. The 5′ ends of reverse primers are biotinylated.

## Data Availability

Data are contained within the article.

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
