# Peer review of "Epigenomic Alterations of the Human CYP11B Gene in Adrenal Zonation"

_ijms, 2024, doi:10.3390/ijms252211956_

Round 1
Reviewer 1 Report
Comments and Suggestions for Authors
In this study, Takeda et al. examined zone-specific DNA methylation levels of promoter regions of CYP11B1/B2 genes and the expression patterns that define distinct functional layers of the adrenal cortex. The authors observed that three CpG dinucleotides in the promoter region of CYP11B2 showed relatively low methylation in the ZG compared to ZF or ZR. In contrast, CpG methylation of CYP11B1 showed hypomethylation in the ZF compared to ZG or ZR. The authors then examined DNA methylation patterns of CYP11B2 in the adrenal medulla, adrenal adenomas, and pheochromocytomas and found a subtle reciprocal relationship between DNA methylation and CYP11B2 expression. While the study is somewhat interesting, it lacks robust data that would enable the authors to conclude that the DNA methylation levels may be associated with expression differences. For instance, DNA methylation analysis of CYP11B2 in the three zones shows significant differences. However, without expression data, it is not clear whether these are associated with gene function. The readers are led to infer this relationship based on previous works. Still, it is possible to perform LCM, obtain DNA and RNA from the same samples, and demonstrate this relationship. Also, the study lacks analytical details such as actual values for percent DNA methylation and relative expression. The results section, whether in the text or a table, should display all values.
Major criticism:
Transparency is needed. The authors need to show DNA methylation levels for all CpGs assayed for both gene promoters and for both methods. While the bisulfite PCR followed by colony sequencing is semiquantitative, it has the benefit of providing spatial patterns of DNA methylation for each DNA clone. The authors should have a supplementary table showing all CpG methylation levels for both genes and a supplementary figure showing the colony sequencing results.
Did the authors attempt to extract mRNA and perform gene expression analysis from the autopsy samples? The authors should examine gene expression from the adrenal cortex samples. While it is understood that mRNA degradation is possible depending on the postmortem interval, the authors should make some effort to obtain mRNA.
For CpG methylation of CYP11B1 in Figure 2, which CpG was examined?
For Figure 3, which CpGs for CYP11B2 were examined?
The authors can use the Qiagen RNeasy FFPE Kit to obtain gene expression information on CYP11B2 in the adrenal medulla.
(https://www.qiagen.com/us/products/discovery-and-translational-research/dna-rna-purification/rna-purification/total-rna/rneasy-ffpe-kit)
Minor points
Affiliation #7 is missing for the senior author.
Comments on the Quality of English LanguageSome grammatical errors should be fixed. For instance, in the abstract, "These results indicate the possibility that the synthesis of occurs in the adrenal medulla."
Reviewer 2 Report
Comments and Suggestions for Authors
The study highlights the zone-specific methylation of CYP11B2 promoters in the human adrenal gland. However, the lack of gene expression data (e.g., RNA-seq) for the adrenal cortex which could be evaluated alongside the methylation data, reduces the depth of the analysis. Similarly, in the case of the adrenal medulla (Fig. 3), qPCR results are provided for NFA and Pheo but not for the adrenal medulla, making the dataset incomplete and diminishing the overall impact of the manuscript.
The sex of adrenal samples should be specified and analyzed separately, especially since the adrenal gland is known to be a sexually dimorphic organ.
Figure 2: Add a color key to visually indicate which color represents each zone. Use a dot plot to show individual data points, similar to Figure 3.
Figure 3: GAPDH should be italicized. For CYP11B2 expression in Pheo, the four undetected samples should be distinguishable from the three CYP11B2-expressing samples. Consider using a different color for those four samples or add arrows to mark them as "not detected."
Round 2
Reviewer 1 Report
Comments and Suggestions for Authors
I reviewed the revised version of the manuscript and found that none of the relatively few suggestions have been added. In the response letter, the authors state that they have fixed the small grammatical error in the abstract and added the affiliation of the senior author (7). In addition, they mention that the additional CpG designation has been added and that Figure 3 has been edited. However, the version that I downloaded and checked twice (v2) does not have any of the edits. While the authors have added supplementary data showing the results of the colony sequencing, they did not include the CpG raw values for all of the samples in a table format for the sake of transparency.
I think the authors or IJMS did not upload the proper version of the manuscript. I downloaded both v1 and v2 and saw no differences in the manuscript versions. Please address this. I appreciate that the authors tried their best to obtain expression values from the LCM samples. However, they should address all other issues requested in the critique.
Author Response
Reviewer 1 “I reviewed the revised version of the manuscript and found that none of the relatively few suggestions have been added. I think the authors or IJMS did not upload the proper version of the manuscript.”
Thank you for your comments. I am very surprised that you did not get my revised version. I sent my revised version to the editor. He told me that he sent my revied version to you. I upload my revised version again. I modified or added several sentences which were underlined in the text.
Reviewer 2 Report
Comments and Suggestions for Authors
Without the correlated gene expression data, the methylation profile alone limits the impact of the results, especially since the zonal difference is only around 10%.
Author Response
Reviewer 2 “Without the correlated gene expression data, the methylation profile alone limits the impact of the results, especially since the zonal difference is only around 10%.”
Thank you for your comment. Correlation data may be important. It should be necessary to analyze more samples. Yoshi et al (Hypertension 2016;68:1432-1437) reported that the differences of methylation rate of CYP11B2 between aldosterone-producing adenomas (APAs) and non-functioning adreno-cortical adenomas (NFAs) were from 0.1 to 0.32 and no correlation of mRNA of CYP11B2 between APAs and NFAs were seen. Dalmazi et al (J Clin Endocrinol Metab 2020; 105:e4605-e4615) reported that the difference of methylation rate of CYP11B2 between aldosterone-producing adenomas (APAs) and adjacent adrenal tissues was 0.18 and CYP11B2 expression and DNA methylation were negatively correlated. We made partially methylated CYP11B2 (14% - 29%) and transfected it to cells and found significant decrease of CYP11B2 compared with no methylated CYP11B2 (J Am Heart Assoc. 2018;7:e008281. DOI: 10.1161/JAHA.117.008281). The difference of around 10% of methylation may affect gene expression.
Round 3
Reviewer 1 Report
Comments and Suggestions for Authors
Thank you for sending me the revised manuscript. I appreciate the edits made by the authors and thus feel comfortable endorsing its publication in IJMS.
Comments on the Quality of English LanguagePlease go over the edits carefully to correct any grammar and spelling errors.
For instance, Line 104 should be "using an ISOGEN kit."
Line 220: "epinegetics" should be "epigenetics"
Reviewer 2 Report
Comments and Suggestions for Authors
Figure 2 (right panel): Use a dot plot to display individual data points.
For all figures: Include the mean on the dot plot.
Figure 3: All samples shown in the same figure should be processed using the same methods to ensure the data are comparable. This applies to both qPCR and the methylation ratio. If the adrenal medulla samples differ from the NFA and Pheo samples, making RNA extraction from the adrenal medulla impossible, how can the methylation ratio data from the adrenal medulla be compared to that of the NFA and Pheo samples?
Comments on the Quality of English LanguagePlease separate Figure 3 into panels 3A and 3B, and revise the figure legend to clearly explain each panel. For example, clarify what is meant by 'not detectable' in both the methylation ratio and expression level.